# Finite Element Modeling and Calibration of a Three-Span Continuous Suspension Bridge Based on Loop Adjustment and Temperature Correction

**DOI:** 10.3390/s24175641

**Published:** 2024-08-30

**Authors:** Hai Zong, Xun Su, Jianxiao Mao, Hao Wang, Hui Gao

**Affiliations:** 1School of Transportation, Southeast University, Nanjing 211189, China; 230228350@seu.edu.cn; 2Key Laboratory of Concrete and Prestressed Concrete Structures of Ministry of Education, Southeast University, Nanjing 211189, China; wanghao1980@seu.edu.cn (H.W.); hgao1993@126.com (H.G.)

**Keywords:** three-span suspension bridge, displacement-limiting suspenders, shape-finding, finite element modeling (FEM), static and dynamic analysis

## Abstract

Precise finite element modeling is critically important for the construction and maintenance of long-span suspension bridges. During the process of modeling, shape-finding and model calibration directly impact the accuracy and reliability. Scholars have provided numerous alternative proposals for the shape-finding of main cables in suspension bridges from both theoretical and finite element analysis perspectives. However, it is difficult to apply these solutions to suspension bridges with special components. Seeking a viable solution for such suspension bridges holds practical significance. The Nanjing Qixiashan Yangtze River Bridge is the first three-span suspension bridge in China. To maintain the configuration of the main cable, the suspension bridge is equipped with specialized suspenders near the anchors, referred to as displacement-limiting suspenders. It is the first suspension bridge in China to use displacement-limiting suspenders and their anchorage system. Taking the suspension bridge as a research background, this paper introduces a refined finite element modeling approach considering the effect of geometric nonlinearity. Firstly, based on the loop adjustment and temperature correction, the shape-finding and force assessment of the main cables are carried out. On this basis, a nonlinear finite element model of the bridge was established and calibrated, taking into account factors such as pylon settlement and cable saddle precession. Finally, the static and dynamic characteristics of the suspension bridge were thoroughly investigated. This study aims to provide a reference for the design, construction and operation of the three-span continuous suspension bridge.

## 1. Introduction

Modern suspension bridges are typically the preferred choice for extremely long-span bridges due to their lightweight nature and exceptional spanning capacity. Due to advances in computational theories, high-performance composites, computer technology and construction methods, the spans of suspension bridges are continuously expanding [1,2]. The 1915 Çanakkale Bridge in Turkey holds the record for being the longest suspension bridge ever constructed, boasting a main span of 2023 m [3]. The Zhangjinggao Yangtze River Bridge in China, with a span of 2300 m, is currently the largest suspension bridge under construction globally [4]. There is no doubt that the tensile strength of the main cables of suspension bridges under self-weight or initial tension is the primary factor determining the continuous expansion of the main span [5].

In essence, the gravity stiffness comes from the geometric nonlinearity, which depends heavily on the configuration and internal forces of the main cable. Nonlinear geometrical behavior occurs due to second-order effects from normal forces and external loads as the main cable undergoes large displacements. In addition to the need to develop a nonlinear relationship between forces and displacements, obtaining the initial geometric configuration is a difficult task since a large number of variables need to be identified [6,7]. Many studies have focused on this subject [8]. At present, the methodology for determining the shape of the main cable can be broadly divided into analytical-based methods and finite element modeling (FEM)-based methods. The analysis method is generally based on the segmented parabolic theory and segmented catenary method (SCM) [9,10,11,12]. The configuration of the main cables in the completed bridge state is first calculated and then evaluated during the construction phase according to the principle of equal lengths of stress-free cables. The segmented parabolic theory assumes that the shape of the main cable is parabolic under a uniform load, and this approximation ignores the concentrated forces transmitted by the suspenders [5,13].

The application of large-scale integrated finite element analysis programs has made it possible to analyze structures of various spans and forms. Scholars have developed various shape-finding methods for the main cables of suspension bridges, such as the Targeted Configuration Under Dead Load (TCUD) method [14], the modified TCUD method [15], the generalized TCUD method [16], the coordinate iteration method [17] and the perturbation method [18]. In these methods, the unstrained length of each main cable segment is unknown and is solved in continuous nonlinear equations using nonlinear finite element iterations. Related research has been carried out in recent years, focusing on the optimization and careful analysis of these calculation methods [19,20,21,22]. Zhang et al. [23] proposed an analytical algorithm to estimate the effect of thermal effects on the shape of the main cable, taking into account the tower saddle–arc effect as well as the combined cable–hanger–girder effect. Wang et al. [24] presented an analytical calculation method considering the combined effect of a main cable–vibration damper–girder. Shin et al. [25] developed a deflection theory for ground-anchored suspension bridges with manufactured convexity by considering the tower effect and the large deflection effect of the main cables. These theoretical derivations are computationally intensive, and the implementation of the algorithms requires a high level of programming skills. Moreover, many of the special elements are not widely embedded in the analysis software commonly used in current engineering, which greatly limits the application of general-purpose analysis software to typical engineering problems [5,9,26].

In the absence of theoretical derivation and original programming, it is more practical for the normal bridge engineer to use it directly for general engineering analysis based on the general bridge analysis software developed for determining the reasonable state of suspension bridges with spatial cables [9,26,27,28]. Zhang et al. [9] proposed a finite element method for shape-finding and force assessment of suspension bridges based on completed loop adjustment. Xiao et al. [26] considered the coupling of spatial main cables and hangers, proposing a five-step algorithm without theoretical derivation and primitive planning to determine the reasonable state of spatial cable of suspension bridges. Sun et al. [17] established a four-step methodology for determining a reasonable completed bridge state, focusing on optimizing the tension and shape of all the cables and independently analyzing the nonlinear effects of each load at the service condition. These methods are not universal, and the modeling accuracy for suspension bridges with special elements needs further consideration. In the case of the Nanjing Qixiashan Yangtze River Bridge, the displacement-limiting suspenders were set up at the side spans near the anchorages, with their lower ends anchored to the transition piers. This makes the reasonable completed bridge state of the suspension bridge different from that of normal suspension bridges. Therefore, it is necessary to propose a refined modeling and main cable shape-finding method for suspension bridges with such special elements.

Based on this research background, this paper proposes a relatively clear nonlinear finite element modeling method, which is mainly carried out by the commercial FE software ANSYS 2022 R1, to provide an optimal solution for the target configuration of a similar suspension bridge. In the proposed method, all the components are included, and the pre-arching of the main cable and pylons is assumed to be adjustable. Firstly, based on the asymptotic criterion and temperature correction, the shape-finding and force assessment of the main cables are carried out. On this basis, the model was calibrated by taking into account factors such as pylon settlement and cable saddle precession. Finally, the static and dynamic characteristics of the suspension bridge were thoroughly investigated by taking the Nanjing Qixiashan Yangtze River Bridge as an example.

## 2. Main Cable Shape-Finding and Modeling Approach

### 2.1. Iterative Calculation of Tension Force and Shape of the Main Cable in the Mid Span

Determining the reasonable state of the mid-span main cable is the most critical step in the algorithm. The mid-span main cable has a well-defined rise-to-span ratio, and the general idea of shape-finding is to solve for stress-free shapes that satisfy the geometric convergence objective under bridge loading. With the powerful geometric nonlinearity function of Ansys, this module is based on the loop adjustment, and the flow chart is shown in Figure 1. The main cables are usually assumed to be flexible rods subjected to tension only. Two endpoints and a center point are used to control the shape, which does not change with the nonlinear iterative process. Then, longitudinal displacement constraints are added at the two endpoints, and the remaining nodes can be modeled initially based on parabolic or catenary curve interpolation. The self-weight of the suspenders, the weight of the cable clamps and the tension of the suspenders are applied as a concentrated load on the IP points of the main cable. The initial strains of the elements are given by Δ/*l*, where Δ is the difference between the length of the element, *l*, and the length of the zero strain. The initial strain reflects the internal stresses of the element at the modeled length, as shown in Equation (1).
(1)$$\varepsilon_{0}=\frac{\sigma_{0}}{E}=\frac{N_{0}}{EA}$$
where *ε*_0_ is the initial strain, *σ*_0_ is the internal stress of the element under the modeling length and *N*_0_ is the internal axial force of the element under the modeling length.

For the dynamic correction factor *β*, which combines the target state and the iterative process, it is calculated according to the following equation.
(2)$$\beta_{n}=\beta_{n-1}\times \frac{∆Y}{d_{y}}$$
where ∆Y is the difference between the resultant y-coordinate of this iteration and the design y-coordinate of the center point, $d_{y}$ is the displacement of the center point of this iteration and the initial *β* needs to be preset, which can be taken as 0.02 based on experience.

After the correction factor is updated, the main cable strain (axial force) needs to be updated accordingly. This ensures that the initial strain correction is larger when the current structure is far from the target state and smaller the closer it is to the target state, which, in turn, leads to faster convergence. Then, return and rerun the nonlinear iteration until the x and y displacement values satisfy the convergence criterion, i.e., less than 0.01 mm, at which point the mid-span shape-finding is completed. The suspender force participates in the overall cycle as one of the discriminating criteria. Subsequent application of the suspender force is taken from the data parameter file corrected. The mid-span shape-finding is used to obtain the stress-free line shape corresponding to the completed bridge state, which also includes the position information of the corresponding cable clamps. The output horizontal force of the main cable is used for side-span main cable shape-finding as a discriminating condition for the equal horizontal force of side-span and mid-span main cables in the completed bridge state. The output vertical force at the IP points is used for the vertical pre-throw height of the pylons to ensure that the position of the saddle meets the error requirements of the bridge criterion.

### 2.2. Iterative Calculation of Tension Force and Shape of the Main Cable in the Side Span

The side-span main cables do not have a predicted rise relative to the mid span, but there is a definite horizontal force of the main cables on the basis of the mid-span shape-finding. The fundamental objective of shape-finding for the side span is to solve the stress-free length so that it can meet the equilibrium conditions with the mid-span cable under the bridge load and, at the same time, meet the geometric convergence target. The general idea of a stress-free length solution in this study is to iteratively search for temperature corrections to satisfy the convergence criteria. The change in length of the main cable following a change in temperature can be calculated using the linear thermal expansion equation:(3)$$\Delta L=\alpha *L_{0}*\Delta T$$
where Δ*L* is the length change, *α* is the coefficient of linear expansion of the main cable, *L*_0_ is the original length and Δ*T* is the temperature change.

The apparent stress-free state of the main cable at the side span is the initial configuration of the model or the corrected model after the procedural solution; however, this does not reflect the true stress-free length, which contains temperature corrections to satisfy the convergence criterion. The temperature is a correction value introduced by the programmed simulation of the unstressed length, which does not exist in the real situation. For the side span, the unstressed length accounted for by the procedure is objectively increased, and the gravity of the structure is increased accordingly. Therefore, the density of the main cable is corrected according to the principle of quality conservation based on the temperature correction value to ensure the high accuracy of the calculation results. The flowchart for shape-finding and force-finding for side-span main cables is shown in Figure 2.

When the main program calls the module for shape-finding of side-span, the temperature-limit values, i.e., *T*_min_ and *T*_max_ in the flowchart, need to be attached to improve the efficiency of the search. The values are related to the initial conformation and temperature sensitivity of the main cables, which are difficult to grasp accurately during the model construction process. Excessive limits will increase the number of cycles, so a cycling function that automatically adjusts the temperature limit is also added to the module. In addition, the nonlinear iterative process of the side span proposed in this paper essentially simulates the true unstressed length through additional temperature corrections, and the initial conformation of the main cable cannot be adjusted. The model correction is only an adjustment of the cable clamp nodes along the initial configuration to meet the geometric accuracy requirement of 0.01 mm.

Theoretically, the initial configuration can be established according to a straight line. For the three-span continuous suspension bridge, the side-span vectors are obvious. The straight-line initial configuration increases the difficulty of convergence to some extent, and the initial configuration can be performed according to a parabola close to the actual vector. The search is performed based on the dichotomy method for expression of the temperature, which is not economical and has a slow convergence rate. The incremental linear interpolation-based method rapidly approaches the target temperature and is extremely fast to solve.

### 2.3. Iterative Calculation of Tension Force and Shape of the Main Cable in the Anchorage Span

In this subsection, only the anchor span and the cable saddle are considered, and an accurate shape-finding is performed for the anchor span. The Spar (or Truss) unit is used to simulate the saddle, and a very large modulus of elasticity is preset to simulate the very high compressive stiffness of the saddle. The general idea of anchor span shape-finding is similar to that of side span, which is to simulate the real stress-free length based on the apparent stress-free model with additional temperature correction. To satisfy the equilibrium condition for a three-span continuous suspension bridge under dead load, the horizontal component of the main cable force in the two anchor spans is equal to that in the side spans. By defining the initial main cable shape in the anchor span as a straight line, the balanced shape can be obtained in the same way as for the side spans. The problem is solved according to the flowchart shown in Figure 2.

After the implementation of the above procedure in a three-span continuous suspension bridge, preliminary main cable shape-finding was achieved. The coordinates and strains, stress-free lengths and forces within the finite element model were extracted from the last modeling of the end of the loop. However, an overall finite element model of the whole bridge still needs to be constructed as the model has been constructed to take into account the effects of each individual member and has been significantly simplified for the pylons and saddles. Previous results will be imported to verify the accuracy of the main cable shape-finding.

## 3. Finite Element Model Calibration

### 3.1. Calibration of Pylon Model

The pylons are only subjected to the gravity and the vertical force transmitted by the main cable in the completed bridge state. The gravity is automatically solved by the model, and the main cable transfer vertical force is the sum of the vertical forces at the endpoints returned during the main cable shape-finding process described above. As a result, the unstressed pylon model under the completed bridge state needs to be established. The pre-throw height is simulated by temperature increasing to overcome the vertical compression generated by gravity and the vertical load to meet the requirements of the IP point error of the saddle. For three-span continuous suspension bridges, the maximum and minimum bending moments of the pylons in the operational condition are not much different, and the force is more balanced. In the completed bridge state, the pylons have zero bending moments in the longitudinal direction, but the need to preset the bending moment cannot be excluded.

For single-span suspension bridges, the maximum and minimum bending moments of the pylons are extremely unbalanced in the operating condition. In this case, it is an economically feasible strategy to control the bending moment in the longitudinal direction of the pylons by applying it in the completed bridge state. There are two ways to apply the pylons to control the bending moment in the longitudinal direction, and the difference is whether the pylons are kept in a vertical position. If the control target is vertical, the pylons must be pre-offset, and the value and law of the pre-offsetting are determined by the stiffness of the pylons and the bending moment of the control target. It is necessary to redefine the side-span shape-finding of the main cable. The application of the control moment in the longitudinal direction of the pylons in the completed bridge state is essentially the control of the unbalanced horizontal force Δ*F* of the side- and mid-span main cable. In the process of shape-finding of the main cable at the side span, it is sufficient to modify the convergence target value. The pylon model needs to be pre-offset according to the induced deformation. In the case of pre-deflection of the pylons of the bridge control target, the initial model of the pylons does not need to be corrected, and only the value of horizontal displacement at the top of the pylons needs to be extracted for the correction of the main cable shape-finding. The control bending moment at the base of the pylons is determined, the unbalanced force on the main cable at the top of the pylons is also well-defined, and the horizontal displacement Δ*X* determined by the stiffness of the pylons can be obtained. The spans of the side- and mid-span main cables are essentially changed, and the effect of Δ*X* needs to be superimposed during shape-finding. The flowchart of the pylons’ modeling procedure is shown in Figure 3. After applying gravity, pre-throw height and Δ*F* loads, the force is connected with the main cables and the Δ*F* loads are removed from the pylons.

### 3.2. Modelling Framework for the Three-Span Continuous Suspension Bridge

The core idea of Ansys-based suspension bridge modeling is to establish a nominal stress-free state model, and after the construction process until the bridge load condition, the various geometric and internal forces meet the requirements of the completed bridge state. The reason why it is called a nominal stress-free state model is mainly based on the technical means (e.g., side-span temperature corrections) used in the modeling process. This enables the unit lengths to have a temperature correction term superimposed on the unstressed state. The aforementioned content introduces the basic process of modeling. Since the iterative shape-finding and solution are completely based on the finite element method, the process is relatively simple, and the result is intuitive, which is easier to be accepted and understood by engineers, and the problems in the solution process are easier to be found and understood [9,27]. The stress modeling process is given in Figure 4. The whole process is based on the structural stress characteristics and properties of the suspension bridge. The solution and shape-finding, however, are prepared according to modularity and are, therefore, generalizable.

The accurate computational analysis and construction control of long-span suspension bridges using Ansys requires research to solve the following problems: a. establishment of a stressful model of the completed bridge; b. nonlinear solution of live load and c. calculation of stress-free length and construction control. This study aims to improve the efficiency of determining the reasonable state of the long-span suspension bridges by simplifying the workflow of the algorithm and reducing the theoretical derivation under the premise of necessary analysis accuracy. Based on the traditional finite element analysis, the algorithm is generated by supplementing several simple operations and process controls. The algorithm can take into account geometric nonlinearities as well as coupled degrees of freedom of the main cables, pylons and girder nodes, and is particularly valuable for shape-finding and force-finding in suspension bridges with spatial cables. The shape-finding sequence is carried out in the order from the center span to the side span and then to the anchor span. Firstly, the horizontal force component of the main cable axial force can be obtained by shape-finding of the mid span. Next, the reasonable state of the side span is analyzed iteratively based on the principle that the horizontal component of the main cable axial force of the two side spans under static loading is equal to that of the mid span. Finally, the anchor spans are analyzed based on the balance of the figure-of-eight saddle. To avoid numerical oscillations or divergence phenomena, the modeling process generally follows the iterative solution idea until the target accuracy is satisfied.

## 4. Case Study

### 4.1. The Nanjing Qixiashan Yangtze River Bridge

The Nanjing Qixiashan Yangtze River Bridge is an asymmetric three-span continuous suspension bridge with a main span of 1418 m, and the distance between the IP points of the north and south anchorages is 2476 m. To coordinate the vertical deformation of the girder and the main cable and to reduce the pressure on the bearing of the transition pier, displacement of the main cable is limited by displacement-limiting suspenders at the transition pier locations [1,29]. Figure 5 illustrates the span arrangement of the suspension bridge. The girder is a fully welded, flat, streamlined closed steel box girder with a top width of 38.8 m (including wind nozzles) and a girder height of 3.51 m at the centerline, and the section is shown in Figure 6. The pylon is in the form of a hybrid structure with a concrete material for the main structure and the optimized shaped rigid superstructure (Figure 7). To reduce the wind resistance coefficient and improve the vortex-induced vibration performance, the pylon cross section was notched with a notch size of 1.6 m × 1.0 m. The cross-sectional characteristics of the girders and the pylons are shown in Table 1. The main cable is made of prefabricated parallel wire strands (PPWS). The cross-sectional area and material properties of the main cable and suspenders are shown in Table 2.

In the target geometric state of the bridge, the structure is modeled with stresses. It is difficult to determine the stress-free state of the structure when the structural stresses are unknown [9]. The key to solving suspension bridges using the finite element method is to know the stress state of each structural member in the completed bridge state. The completed bridge state includes geometric states, such as IP point position at reference temperature, rise span ratio, suspender attitude, etc. and some internal force states, such as zeroing of the bending moment at the bottom of the pylons.

The Nanjing Qixiashan Yangtze River Bridge was modeled and analyzed using ANSYS 2021 R1, and the finite element model is shown in Figure 8. According to that discussed in the framework, the modeling details are as follows:①The main cables and slings are constructed using the LINK10 truss element, whose cross-sectional characteristics are shown in Table 2.②The pylons and girders are constructed using spatial beam element BEAM44, with the cross-sectional areas and material properties shown in Table 1.③The hinged spring, elastic support and longitudinal displacement-limiting between the pylons and the girders are modeled using COMB1N14.④The MASS21 element is adopted in the dynamic calculations to simulate the rotational mass moment of inertia of the girders.⑤The girders are coupled to the suspenders through a rigid connection, and the sagging effect of the suspenders is neglected. All boundary conditions in the model are summarized in Table 3.⑥The model has a spatial rectangular coordinate system in which the *X*-axis is defined as the longitudinal direction of the bridge, the *Y*-axis as the vertical direction, and the *Z*-axis as the transverse direction of the bridge.

**Figure 8 sensors-24-05641-f008:**
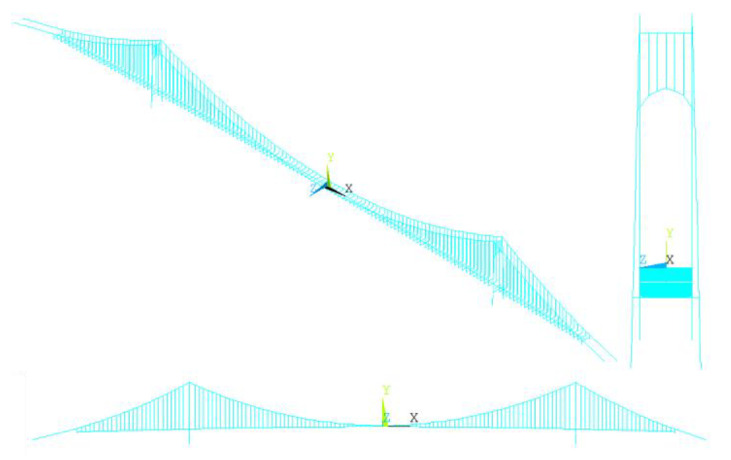
Finite element model of the Nanjing Qixiashan Yangtze River Bridge.

**Table 3 sensors-24-05641-t003:** Boundary conditions.

Position	*D_x_*	*D_y_*	*D_z_*	*R_x_*	*R_y_*	*R_z_*
Anchor node of the main cable	Fixed	Fixed	Fixed	Fixed	Fixed	Fixed
Bottom of the pylon and abutment	Fixed	Fixed	Fixed	Fixed	Fixed	Fixed
The intersection of girder and pylon	Free	Spring-Damper	Fixed	Free	Free	Free
IP point and top of the pylons	Coupled	Coupled	Coupled	Free	Free	Free
Splay saddle	Free	Coupled	Coupled	Free	Free	Free
Girder at abutment	Free	Fixed	Fixed	Fixed	Free	Free
Anchorage node of displacement-limiting suspenders	Free	Spring-Damper	Free	Free	Free	Free

The approach introduced above is used to iteratively calculate the model of the Qixiashan Yangtze River Bridge. The calculation accuracy mainly depends on the solution accuracy, unit density and convergence error. The accuracy of the nonlinear solution is extremely high for Ansys. The convergence error can be controlled artificially. The unit density had little effect on either the internal forces or the lineaments. Table 4 shows that the unit density varies from 1 to 10 m, and the resulting unstressed length of the mids pan has an effect of only 0.2 mm. Large unit lengths (e.g., only one unit between cable clamps) do not accurately simulate the deformation of the main cable between cable clamps by gravity, and the error can be up to about 1 cm. For the scale of Nanjing Qixiashan Yangtze River Bridge, the unit density is controlled to be around 3 m, and the stress-free cable shape obtained is accurate and reliable based on the nonlinear solution. The real shapes and internal forces of all elements in the bridge model are calculated iteratively to obtain the complete solution. The control of the internal forces in the displacement-limiting suspenders is also included in this study to ensure that they are always in tension under any operating condition. To better balance the analysis accuracy and algorithm efficiency, based on multiple tests, it was determined that the convergence conditions satisfied are shown in Table 5.

### 4.2. Analysis of the Reasonable Completed Bridge State

The reasonable completed bridge condition mainly provides a reference for the bridge monitor to track the calculation and error adjustment from the structural construction to the final completed bridge. The goal of monitoring is the designed state of completion. If the target state of the bridge is not consistent, then the monitoring is off course. The iterative computational model in this section is the constant load state computational model of Section 3.2. After repeated iterations and adjustments, the coordinates of the main cable under the dead load state of the bridge are shown in Figure 9. The initial coordinates are calculated from the segmental catenary [10,11]. The results of the acceptance load test of the Qixiashan Yangtze River Bridge were compared and analyzed, and it was found that the coordinates of the main cables obtained from the simulation were basically the same as the measured coordinates of the completed bridge. This validates the effectiveness of the methodology proposed in this study for modeling. The axial force of the main cable obtained from the final iteration is shown in Figure 10. As the spans of the two side spans are not the same, the axial force of the main cable is not completely symmetrical. Among them, the axial force of the main cable near the south pylon is relatively large, approximately 263.34 × 10^3^ kN. Overall, the distribution of the axial force of the main cable is basically in line with the characteristics of the three-span continuous suspension bridge.

The distribution of suspender forces is shown in Figure 11. As mentioned in Section 4.1, the existence of displacement-limiting suspenders on both side spans of the Nanjing Qixiashan Yangtze River Bridge makes the cable force assignment relatively special. From the results of iterative calculations of the reasonable state, the displacement-limiting suspender force of the south and north spans reached 3875.85 kN and 3912.02 kN, respectively. The special suspender forces on the left and right sides of the north and south pylons reached 4069.33 kN and 4058.83 kN. These groups of suspenders have essentially twice the force of the rest of the suspenders. In addition, the adjacent suspenders of the above groups of suspenders also have a slightly higher tension than the other suspenders. The rest of the general locations have a more even distribution of suspender tension of about 1730 kN. Similarly, a comparative analysis was conducted between the simulated cable tension and the measured values from the load test. It is evident that the disparity between the two is negligible, both falling within a margin of 0.5%.

In common engineering practices, displacement is generally measured by the deviation between a reasonable state obtained from iterative calculations and a desired position obtained through nonlinear finite element analysis. The determined state is reasonable when the deformations of all units obtained from the calculations are within the specified tolerances [27]. The displacement deviations obtained for the girder at a reasonable state are shown in Figure 12. The maximum vertical displacement of the girder is close to 2 mm, and the maximum lateral displacement is about 6 mm. It is shown that the dead load displacements are comparatively low, and the structure is able to reach the designed bridge condition under this set of suspender forces. The state found by the algorithm is very close to the ideal position in normal engineering practice. This fact verifies the accuracy and effectiveness of the proposed algorithm.

In the completed bridge state, the internal forces of the beams are shown in Figure 13. The maximum bending moment of the girder is 35,670 kN·m, which occurs at the centerline of the pylon. The maximum shear occurs at the joints between the girder segments in the non-cabled area and the cabled area at the pylons and at the location of the joints in the girder segments in the general cabled area, where the maximum shear is about 320 kN. It can be seen from the stress diagram in Figure 13c that the stresses in the girders in the bridge-forming condition are slight. For the general location, the maximum stress at the upper edge is less than 5 MPa, and that at the lower edge is less than 8 MPa. The maximum stress occurred in the girder at the special suspenders near the pylons, with the maximum compressive stress of 32.54 MPa occurring at the upper edge and the maximum tensile stress of 22.87 MPa occurring at the lower edge.

### 4.3. Comparative Analysis of the Dynamic Characteristics

Based on the static analysis of the aforementioned three-span continuous suspension bridge, further analysis was conducted on the dynamic characteristics of the suspension bridge. The frequencies and mode shapes of the bridges derived from the finite element model are presented in Table 6 and Figure 14, respectively. The bridge exhibits predominant frequencies in the range of 0.1 Hz to 0.22 Hz, with low-order modes dominated by lateral and vertical bending vibrations and no occurrence of vertical flutter. Furthermore, the ambient vibration test caused by excitation of random loads (e.g., wind loads, ground pulsations, water currents, etc.) in the absence of traffic loads, and acceleration data extracted from the Nanjing Qixiashan Yangtze River Bridge SHM system on February 6th, 2020 (Figure 15) were used for the comparative analysis of the dynamic characteristics of the bridge. Comparison of the finite element simulation results with the ambient vibration test and monitoring results of the bridge is shown in Table 6, and the maximum error is only 3.2%, 1.05%. Apart from the impact of operational conditions on the monitoring data, it can be observed that the dynamic characteristics identification results of the three are relatively consistent, further confirming the reliability of the finite element model.

## 5. Conclusions

A case study about modeling, shape-finding and model calibration of a three-span continuous suspension bridge based on nonlinear finite element analysis is presented in this paper. The detailed internal forces and stresses of the components in the dead load are calculated. Because iterative solving relies entirely on finite element methods, there is no need for theoretical derivation. This process is straightforward, yielding intuitive results that are more readily accepted by engineers. Identifying and comprehending the issues that arise during the process of seeking solutions becomes more facile. This approach is applicable to general engineering practices, particularly providing an efficient and convenient method for the preliminary design stage and comprehensive analysis of bridge engineering. The main summaries are as follows:(1)This approach eliminates the need to have prior knowledge of the tension in the shape-finding. It avoids the errors introduced by the use of approximations in conventional methods of calculating and analyzing suspender tensions.(2)The iterative calculation process is significantly affected by the initial transverse coordinates of the main cable nodes near the midpoint. It would be preferable to assume a more optimal initial shape of the main cable near the midpoint of the span, as this would help enhance the efficiency of the iterative calculations.(3)All the components are included in the proposed approach, and the pre-arching of the main cable and pylons is assumed to be adjustable. Therefore, the calculation results are in accordance with engineering practice.(4)Due to the presence of displacement-limiting suspenders, the configuration of the main cable and the distribution of suspender forces in this suspension bridge differ from those of conventional three-span continuous suspension bridges. Therefore, it is essential to employ specialized maintenance and operation measures to ensure its long-term functionality.

## Figures and Tables

**Figure 1 sensors-24-05641-f001:**
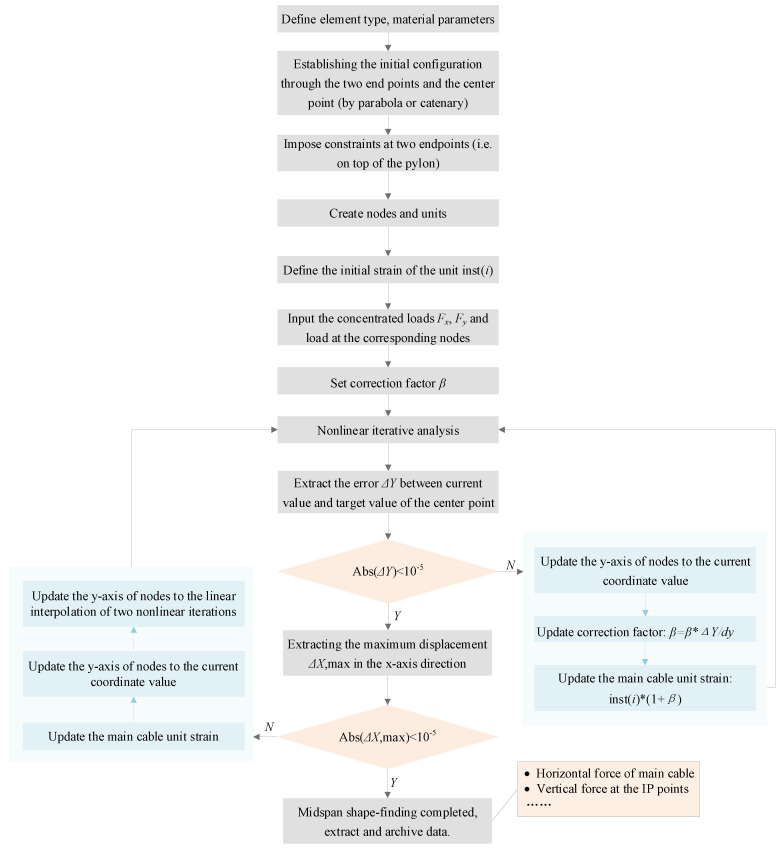
The flowchart of shape-finding and force-finding in the midspan.

**Figure 2 sensors-24-05641-f002:**
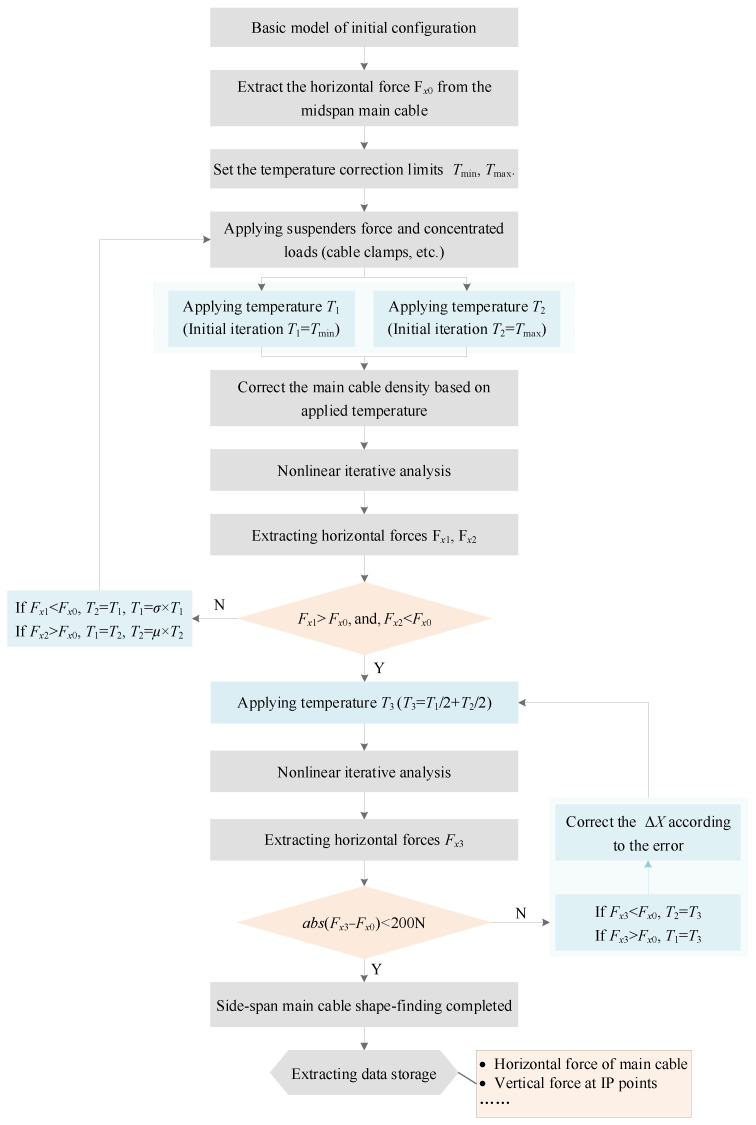
The flowchart of shape-finding and force-finding in the side span.

**Figure 3 sensors-24-05641-f003:**
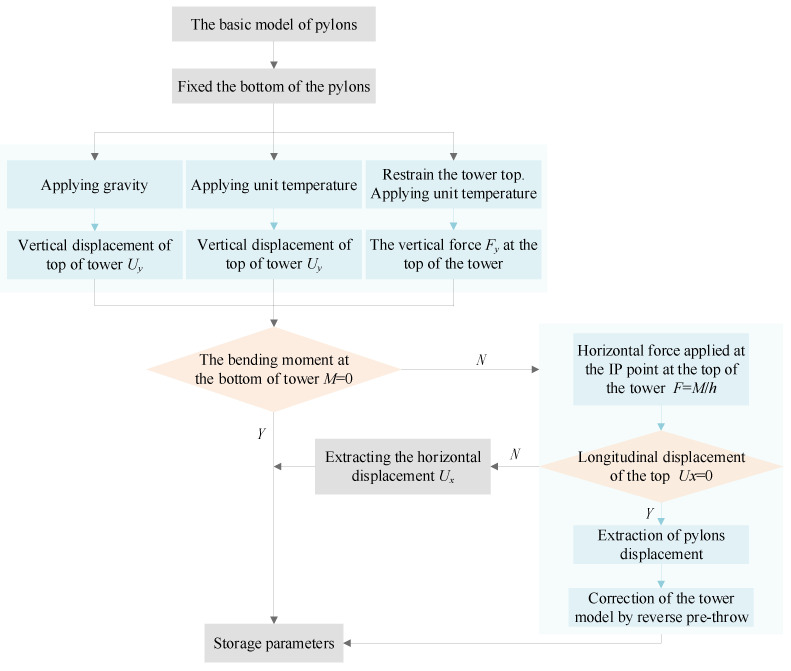
The flowchart of the pylons modeling procedure.

**Figure 4 sensors-24-05641-f004:**
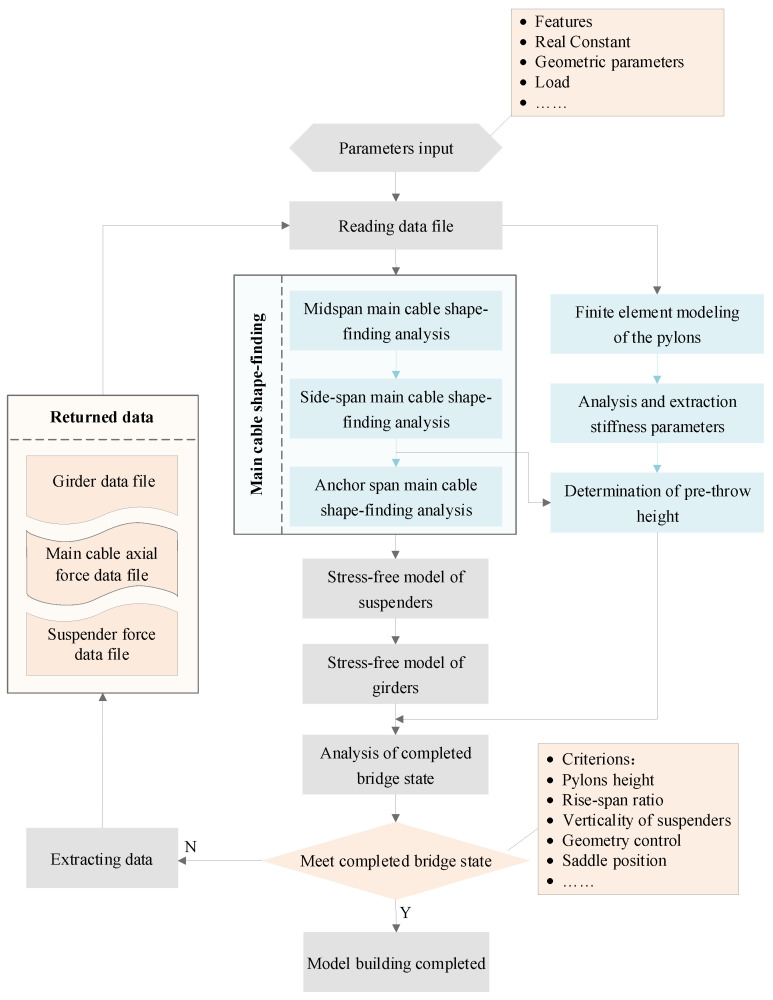
The flowchart of stress modeling for completed bridge state.

**Figure 5 sensors-24-05641-f005:**
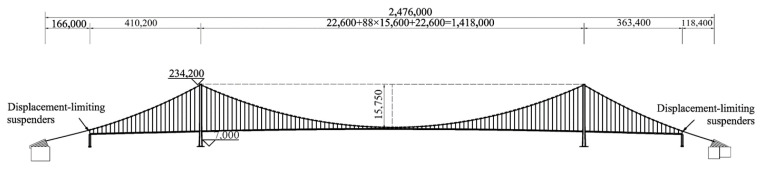
The elevation view of the Nanjing Qixiashan Yangtze River Bridge.

**Figure 6 sensors-24-05641-f006:**
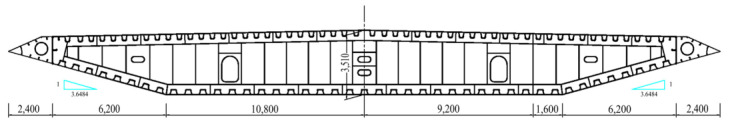
The cross-section of the girders.

**Figure 7 sensors-24-05641-f007:**
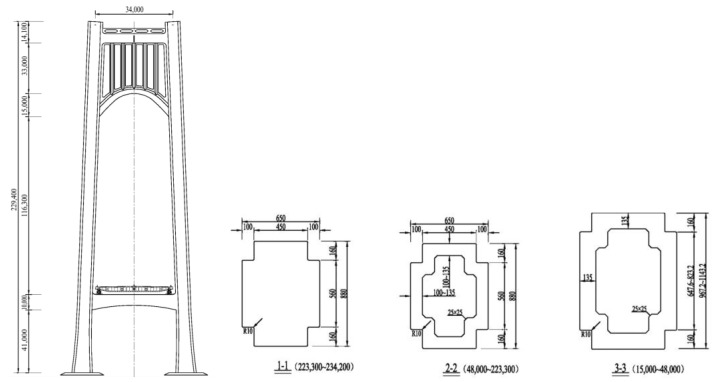
Side view of the pylon and dimensions of each cross-section.

**Figure 9 sensors-24-05641-f009:**
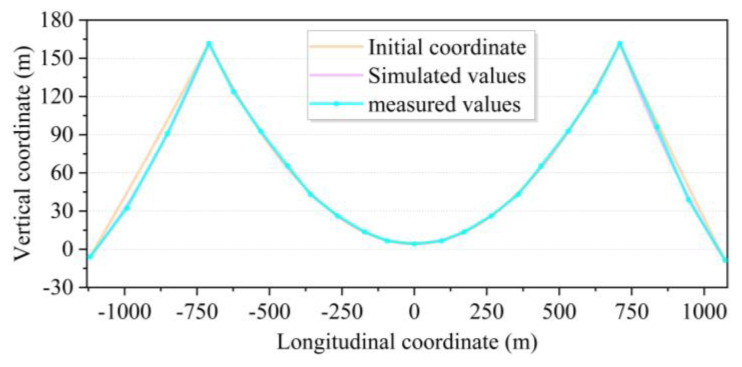
Comparative analysis of simulation results and measured values of main cable coordinates.

**Figure 10 sensors-24-05641-f010:**
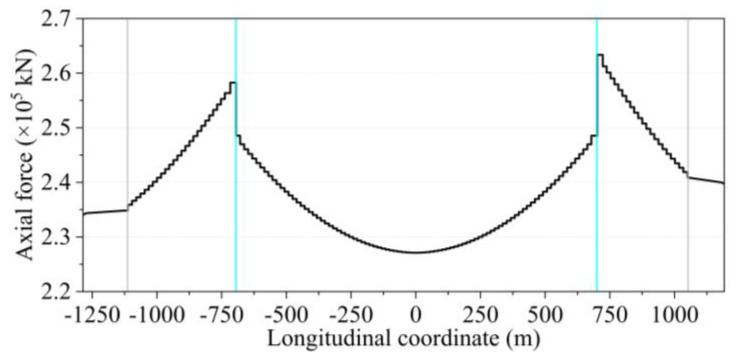
Results after iteration of the main cable axial force.

**Figure 11 sensors-24-05641-f011:**
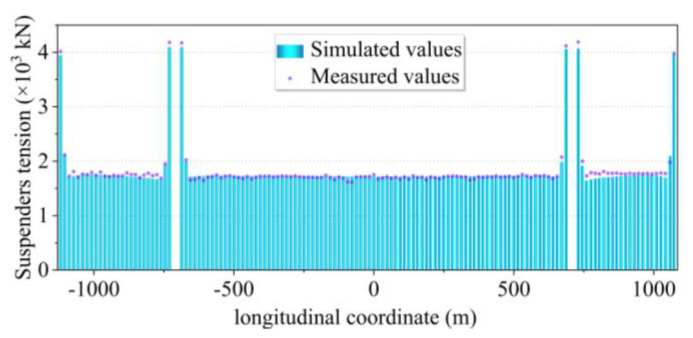
Distribution of detailed suspender forces.

**Figure 12 sensors-24-05641-f012:**
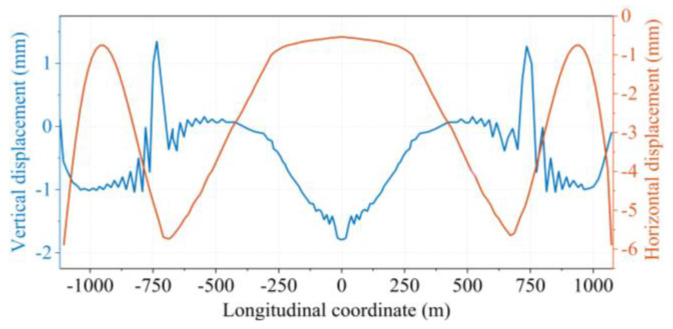
Displacement of girders under dead load at completed bridges.

**Figure 13 sensors-24-05641-f013:**
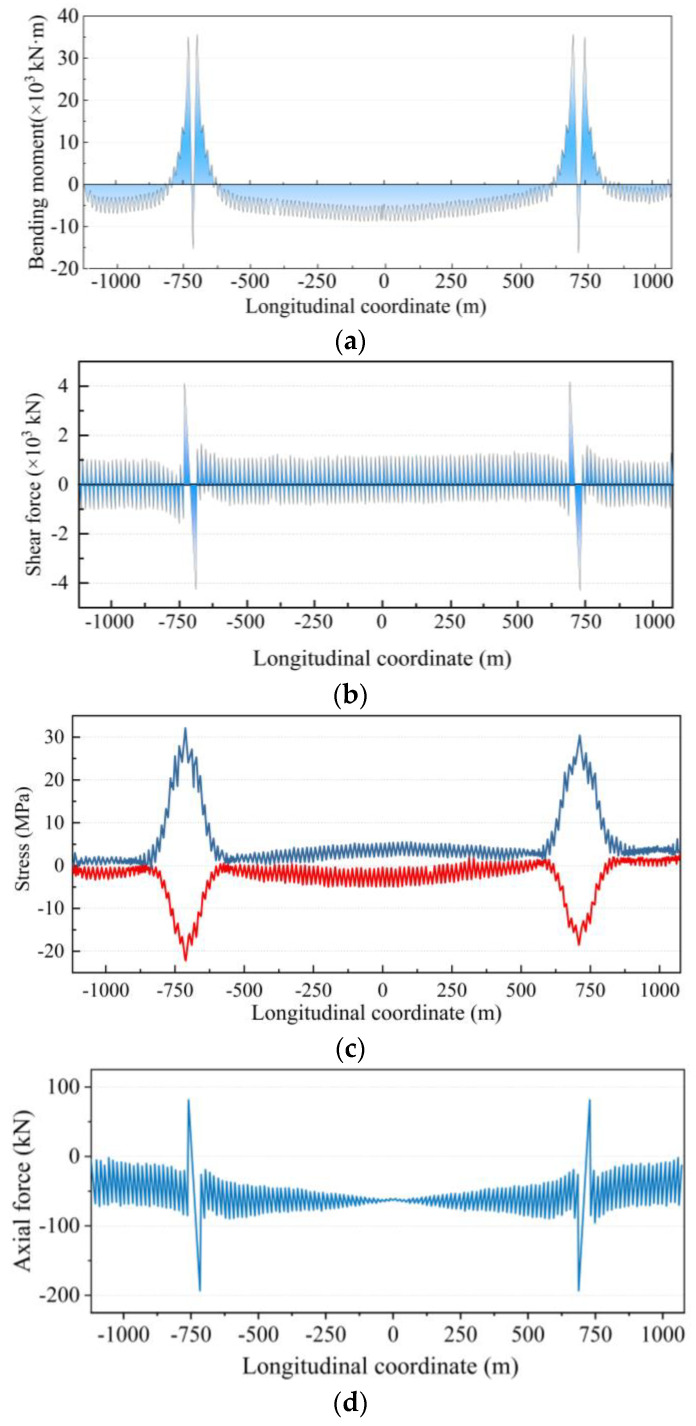
Internal force diagram of girder under the reasonable state: (**a**) the bending moment of the girder; (**b**) the shear force of the girder; (**c**) the stresses of the girder and (**d**) the axial force of the girder.

**Figure 14 sensors-24-05641-f014:**
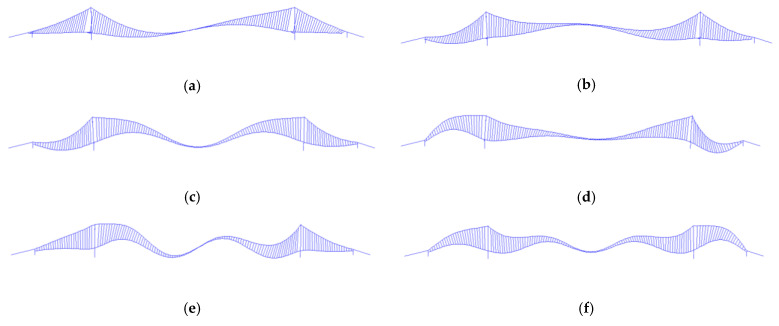
The mode shapes of the bridges derived from the finite element model (**a**–**f**).

**Figure 15 sensors-24-05641-f015:**
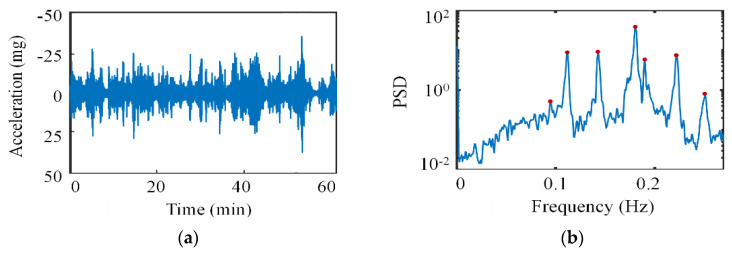
Time and frequency domain analysis of typical acceleration samples based on field monitoring (**a**) time history curve and (**b**) PSD.

**Table 1 sensors-24-05641-t001:** Cross-sectional properties of the girder and pylon.

Cross Section	*A* (m^2^)	*H* (m)	*I_y_* (m^4^)	*I_z_* (m^4^)	*J* (m^4^)
Girder	1.432	/	2.921	155.823	8.064
Pylon (Bottom)	104.86	15.00~48.00	674.58	1181.90	1551.3
Pylon (Middle)	26.72~47.01	48.00~223.30	121.40~388.73	212.67~693.21	256.71~854.72
Pylon (Top)	50.791	223.30~234.20	152.37	284.65	353.8

Note: *A* is cross-section area; *H* is the height of the section; *I_z_* is moment of inertia to the *z*-axis; *I_y_* is moment of inertia to the *y*-axis and *J* is torsion resistance.

**Table 2 sensors-24-05641-t002:** Cross-sectional area and material properties.

Components	Position	Cross-Sectional Area (mm^2^)	Tensile Strength (MPa)	Elastic Modulus (MPa)	Safety Factor
Main cable	North span	4.025 × 10^5^	1770	2 × 10^5^	2.5
	Mid span	3.854 × 10^5^			
	South span	4.082 × 10^5^			
Suspenders	Common	4.04 × 10^3^	1670	2 × 10^5^	2.5
	Special	8.29 × 10^3^			
	Displacement-limiting	2.27 × 10^4^			

**Table 4 sensors-24-05641-t004:** Stress-free length difference in the mid span of the main cable.

Unit Density	Stress-Free Length of the Main Cable	Unit Density	Stress-Free Length of the Main Cable
1 m	1459.07183 m	5 m	1459.07180 m
3 m	1459.07182 m	10 m	1459.07159 m

**Table 5 sensors-24-05641-t005:** Convergence determination conditions of nonlinear iterative analysis.

Modules	Discriminant	Admissible Error
Shape-finding of mid-span main cable	Rise	Δ*f* < 0.01 mm
	Cable clamp position	Δ*X* < 0.01 mm
Shape-finding of side-span main cable	Difference with mid span on horizontal force	Δ*F_x_* < 10 N
	Cable clamp position	Δ*X* < 0.01 mm
Calculation of completed bridge state	Pylon height	Δ*H* < 0.1 mm
	Rise at mid span	Δ*f <* 0.01 mm
	Cable clamp position	Δ*X* < 0.01 mm
	Girder geometry	Δ*X*, Δ*Y* < 0.01 mm
	Bending moment at the bottom of the pylons	*M* = 0
	IP point of saddle	Δ*X*, Δ*Y* < 0.01 mm
	The difference between output and input suspender forces	Δ*F* < 200 N

**Table 6 sensors-24-05641-t006:** Dynamic characteristics of Nanjing Qixiashan Yangtze River Bridge.

Numbers	FEM/Hz	Monitoring Results/Hz	Ambient Vibration Test/Hz	Modal Shapes
1	0.1033	0.105	0.11	1st antisymmetric vertical vibration
2	0.1150	0.116	/	1st symmetric vertical vibration
3	0.1462	0.147	0.14	2nd symmetric vertical vibration
4	0.1801	0.181	0.18	Side-span antisymmetric vertical vibration
5	0.1952	0.189	0.19	2nd antisymmetric vertical vibration
6	0.2236	0.224	0.22	symmetric vertical vibration

## Data Availability

The data presented in this study are available upon request from the corresponding author. The dataset and code cannot be shared due to specific reasons.

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
