# Peer review of "Finite Element Modeling and Calibration of a Three-Span Continuous Suspension Bridge Based on Loop Adjustment and Temperature Correction"

_sensors, 2024, doi:10.3390/s24175641_

Round 1

Reviewer 1 Report

Comments and Suggestions for Authors

This manuscript proposed a refined finite element modeling approach considering the effect of geometric nonlinearity to investigate the suspension bridge. This work should be carefully revised before it can be considered for publication. The specific comments are listed below:

(1) The novelty of this works should be further stated in Introduction since it was mainly conducted on the platform of commercial FE software ANSYS.

(2) Four flowchart were listed in the manuscript. However, those flowchart were badly designed. Please consider to re-organize the flowcharts to make them more concise and easy to understand.

(3) The element and node numbers of the FE model shown in Fig. 8 are missing. Did the authors conduct mesh density check?

(4) The location of the accelerometers in the bridge SHM system should be given in detail. How did the authors identify the 7th order frequency from Fig. 14 without checking the corresponding mode shapes? The first 7 mode shapes should be given in the manuscript.

Comments on the Quality of English Language

There are some grammar and typo errors in the manuscript. Please check the manuscript carefully and revise them.

Author Response

Comment #1: There are some grammar and typo errors in the manuscript. Please check the manuscript carefully and revise them.

Response: Thanks for pointing out the English written problems. We have carefully checked the manuscript. All the grammatical and professional vocabulary have been corrected and marked by red font.

Comment #2: The novelty of this works should be further stated in Introduction since it was mainly conducted on the platform of commercial FE software ANSYS.

Response: Thanks for the suggestion. The author has made modifications to the Introduction section, mainly in the last two paragraphs of the section. Following the advice provided by the reviewers, the authors have highlighted in the Introduction that this study provides a convenient method of shape finding and modelling mainly for suspension bridges with limiting suspension cables, with all members included and assuming adjustable pre-arches for the main cables and towers. And all these steps are basically carried out on commercial FE software ANSYS. This makes it easier for engineers to understand and accept. (Page 2 Line 89-104)

Based on this research background, this paper proposes a relatively clear nonlinear finite element modelling method, which is mainly carried out by the commercial FE software ANSYS, to provide an optimal solution for the target configuration of a similar suspension bridge. In the proposed method, all the components are included and the pre-arching of main cable and pylons is assumed to be adjustable. Firstly, based on the asymptotic criterion and temperature correction, the shape finding and force assessment of the main cables are carried out. On this basis, the model was calibrated by taking into account factors such as pylon settlement and cable saddle precession. Finally, the static and dynamic characteristics of the suspension bridge were thoroughly investigated by taking the Nanjing Qixiashan Yangtze River Bridge as an example.

Comment #3: Four flowchart were listed in the manuscript. However, those flowcharts were badly designed. Please consider to re-organize the flowcharts to make them more concise and easier to understand.

Response: Thanks for the comments. Based on the suggestions of the reviewers, the authors have made appropriate modifications to the four flowcharts to make them easier to understand.

Comment #4: The element and node numbers of the FE model shown in Fig. 8 are missing. Did the authors conduct mesh density check?

Response: Thanks for the suggestion. Since the model has a large number of nodes and units, it is not easy to see them in the picture and highlights the clutter of the model, the authors have deleted the descriptions of node numbers and unit numbers in Figure 8 of the manuscript. However, in this study, the authors have fully examined the grid density, and this section is mainly reflected in page 12, lines 350-356 and Table 4. To balance computational efficiency and model accuracy, the cell density is controlled to be around 3m.

Table 4 shows that the unit density varies from 1 to 10 m, and the resulting unstressed length of the midspan has an effect of only 0.2 mm. Large unit lengths (e.g. only one unit between cable clamps) do not accurately simulate the deformation of the main cable between cable clamps by gravity, and the error can be up to about 1cm. For the scale of Nanjing Qixiashan Yangtze River Bridge, the unit density is controlled to be around 3m, and the stress-free cable shape obtained is accurate and reliable based on the nonlinear solution.

Table 4 Stress-free length difference in the midspan of the main cable

Unit density

Stress-free length of the main cable

Unit density

Stress-free length of the main cable

1m

1459.07183m

5m

1459.07180m

3m

1459.07182m

10m

1459.07159m

Comment #5: The location of the accelerometers in the bridge SHM system should be given in detail. How did the authors identify the 7th order frequency from Fig. 14 without checking the corresponding mode shapes? The first 7 mode shapes should be given in the manuscript.

Response: Thanks for the suggestion. The authors have given details of the placement of accelerometers in SHM systems in their previous studies on modal parameter identification [1]. In that study the judgement of the modal vibration shapes was made. The modal frequencies in Fig. 14 are based on the judgement made in that study. In addition, considering the length of the manuscript, the authors have listed in Table 6 the modal vibration shapes corresponding to each order of modal frequency. There may be a slight repetition if the vibration pattern pictures are added. If the reviewer suggests that the modal shape pictures are necessary, the authors will consider adding them in the next revision.

  • Mao, J.X.; Su, X.; Wang, H.; Li, J. Automated Bayesian operational modal analysis of the long-span bridge using machine-learning algorithms. Engineering Structures. 2023, 289, 116336.

Reviewer 2 Report

Comments and Suggestions for Authors

The paper is about FEM and calibration of a long suspension bridge. The following comments should be addressed before acceptance:

1- There are so many errors related to citations in the text that should be corrected.

2- Fig 3 shows the pylons' cross sections but it is not mentioned their locations. It should be clarified.

3- The paper is difficult to follow for readers. It cannot be understood that sections: 2 or 3 presents the methods or the application of the method on the bridge. It is suggested to mention the methods and then present the application of the mentioned method on the bridge. 

4- Fig 8 is not clear enough. More figures of the bridge with better quality should be added to this paper.

5- In Table 6, What is the frequencies related to the load test? it should be clarified. Why do not we have the first frequency value for it?

6- The calibrated model's mode shapes should be presented in section 4.2. The first three fundamental modes are enough.

Author Response

Overall Comment: The paper is about FEM and calibration of a long suspension bridge. The following comments should be addressed before acceptance:

Response: The authors appreciate the reviewer’s comments and valuable suggestions on our study. These comments and suggestions are of great help in improving the overall quality of the manuscript. The authors have carefully revised the manuscript in response to each specific comment.

Comment #1: There are so many errors related to citations in the text that should be corrected.

Response: Thanks for the comments. The authors thoroughly examined and revised all citations in the manuscript. In addition, details such as these in the manuscript were likewise checked and revised to avoid low-level errors.

Comment #2: Fig 3 shows the pylons' cross sections but it is not mentioned their locations. It should be clarified.

Response: Thanks for the comments. The authors have clearly labelled the locations of the three sections as well as their heights in the pylon cross-section in Fig. 3 and in Table 1.

Fig. 7 Side view of the pylon and dimensions of each cross-section

Table 1 Cross-sectional properties of the girder and pylon

Cross section

A (m2)

H (m)

Iy (m4)

Iz (m4)

J (m4)

Girder

1.432

/

2.921

155.823

8.064

Pylon (Bottom)

104.86

15.00~48.00

674.58

1181.90

1551.3

Pylon (Middle)

26.72~47.01

48.00~223.30

121.40~388.73

212.67~693.21

256.71~854.72

Pylon (Top)

50.791

223.30~234.20

152.37

284.65

353.8

Note: A is cross-section area; H is the height of the section; Iz is moment of inertia to the z-axis; Iy is moment of inertia to the y-axis; J is torsion resistance.

Comment #3: The paper is difficult to follow for readers. It cannot be understood that sections: 2 or 3 presents the methods or the application of the method on the bridge. It is suggested to mention the methods and then present the application of the mentioned method on the bridge.

Response: Thanks for the comments. The authors have reorganized and revised the structure of the manuscript, mainly in Section 2, 3 and 4. In Section 2 and 3, the main cable shape-finding and modelling methods are mainly introduced, and the introduction of Nanjing Qixiashan Yangtze River Bridge and the application of the mentioned method on the bridge are put into the Case study of Section 4.

Comment #4: Fig 8 is not clear enough. More figures of the bridge with better quality should be added to this paper.

Response: Thanks for the comments. The authors have modified Fig. 8 and re-uploaded a clearer finite element model of the Nanjing Qixiashan Yangtze River Bridge. (Page 11 Line 327)

Fig. 8 Finite element model of the Nanjing Qixiashan Yangtze River Bridge

Comment #5: In Table 6, What is the frequencies related to the load test? it should be clarified. Why do not we have the first frequency value for it?

Response: Thanks for the comments. The authors have misrepresented in this section that the self-resonance characteristics of the structure were measured by the pulsation test and not the load test. It was tested and analyzed using the DH5907 wireless test system. The authors have corrected this section in the manuscript.

In addition, the first order is missing due to the fact that the test values of the pulsation test do not measure all modes.

Table 6 dynamic characteristics of Nanjing Qixiashan Yangtze River Bridge

Numbers

FEM/Hz

Monitoring results/Hz

Pulsation test /Hz

Modal shapes

1

0.1033

0.105

/

1st antisymmetric vertical vibration

2

0.1150

0.116

0.11

1st symmetric vertical vibration

3

0.1462

0.147

0.14

2nd symmetric vertical vibration

4

0.1801

0.181

0.18

Side-span antisymmetric vertical vibration

5

0.1952

0.189

/

2nd antisymmetric vertical vibration

6

0.2236

0.224

0.22

symmetric vertical vibration

Comment #6: The calibrated model's mode shapes should be presented in section 4.2. The first three fundamental modes are enough.

Response: Thanks for the comments. Considering the length of the manuscript, the authors have listed in Table 6 the modal vibration shapes corresponding to each order of modal frequency. There may be a slight repetition if the vibration pattern pictures are added. If the reviewer suggests that the modal shape pictures are necessary, the authors will consider adding them in the next revision.

Round 2

Reviewer 1 Report

Comments and Suggestions for Authors

Accept

Author Response

We are grateful for the valuable suggestions provided by the reviewers and their acknowledgment of our manuscript.

Reviewer 2 Report

Comments and Suggestions for Authors

The authors addressed the comments and the manuscript was improved but there are some comments that are needed to be addressed in the text.

1- What is a pulsation test? it is just presented at the end of the manuscript without any further description before it. What was the aim of this test? Why could't we find the first natural frqucey from this test?

2- All the 6 mode shapes of the calibrated model should be presented because the terms antisymmetric or symmetric are unclear. Moreover these figures can be more understandble for readers and other researchers who want to perform analysis on these type of bridges. Morever it should be mentioned that the mode shapes were dervied just from the numerical model or from the tests or both.

3- an X-Y front view of the bridge numerical model should be added to figure 8.

Author Response

Overall Comment: The authors addressed the comments and the manuscript was improved but there are some comments that are needed to be addressed in the text.

Response: The authors appreciate the reviewer’s comments and valuable suggestions on our study. We have made revisions according to each of the following detailed comments.

Comment #1: What is a pulsation test? it is just presented at the end of the manuscript without any further description before it. What was the aim of this test? Why could't we find the first natural frqucey from this test?

Response: Thanks for the comment. In the previous revision, the authors incorrectly expressed ambient vibration test as pulsation test. The authors have revised it in their latest manuscript. Bridges are subjected to small and irregular vibrations due to external disturbances (natural winds, earth pulsations, etc.), and the ambient vibration signals of the structure under random environmental excitations are measured and recorded using highly sensitive vibration sensors. Since the ambient vibration signal contains rich frequency components and is a smooth state through a random process, the natural vibration frequency can be obtained directly through the FFT analysis of the ambient vibration signal.

In this study, the ambient vibration test results of Qixiashan Yangtze River Bridge are included only to verify the accuracy of the dynamic characteristic results of the established model, and therefore do not appear in the previous modelling and analysis process. As the process is smooth and random, the first natural vibration frequency of the bridge in this test was not measured, and the results are not shown in Table 6.

Comment #2: All the 6 mode shapes of the calibrated model should be presented because the terms antisymmetric or symmetric are unclear. Moreover these figures can be more understandble for readers and other researchers who want to perform analysis on these type of bridges. Morever it should be mentioned that the mode shapes were dervied just from the numerical model or from the tests or both.

Response: Thanks for the comment. Following the kind suggestion of the reviewer, the authors have added the vibration shapes of the 6-order modes to the manuscript to make it more convenient and easier to understand for the readers and other researchers. In addition, the authors make a point of mentioning in the manuscript that the mode shape were obtained by means of finite element modelling. (Page 17 Lines 440-443)

The frequencies and mode shapes of the bridges derived from the finite element model are presented in Table 6 and Fig. 14, respectively.

Comment #3: an X-Y front view of the bridge numerical model should be added to figure 8.

Response: Thanks for the comment. As suggested by the reviewer, the authors have added a model image of the x-y axis view of the Nanjing Qixaishan Yangtze River Bridge in Fig. 8. In addition, the top view has been censored by the authors to make the image clearer and to avoid repetition.

Fig. 8 Finite element model of the Nanjing Qixiashan Yangtze River Bridge

We tried our best to improve the manuscript and made some changes in the manuscript. These changes will not influence the content and framework of the paper. And we have marked the changes in red in revised paper.

We appreciate for Editors/Reviewers’ warm work earnestly, and hope that the correction will meet with approval.

Once again, thank you very much for your comments and suggestions.

Round 3

Reviewer 2 Report

Comments and Suggestions for Authors

The manuscript was improved but two big and important questions still were not answered.:

1- What is the monitoring results in Table 6? It is not ambient vibration test? How did you perform it? It should be explained in the text and the differences of the processes should be described

2- What do you mean that ^As the process is smooth and random, the first natural vibration frequency of the bridge in this test was not measured, and the results are not shown in Table 6.^ ?

How do you know that 0.11 Hz is not your first frequncy from ambient vibration testing? Did you derive the mode shapes from ambient vibration tests as well?

Author Response

Overall Comment: The manuscript was improved but two big and important questions still were not answered:

Response: The authors appreciate the reviewer’s comments and valuable suggestions on our study. We have made revisions according to each of the following detailed comments.

Comment #1: What is the monitoring results in Table 6? It is not ambient vibration test? How did you perform it? It should be explained in the text and the differences of the processes should be described

Response: Thanks to the commenters' valuable suggestions, the authors have revised the manuscript. The ambient vibration test is used to determine the minor vibration response of the bridge caused by the excitation of random loads such as wind loads, ground pulsation, water flow, etc. at the bridge site without any traffic loads and without regular vibration sources near the bridge site. In this study, a DH5907 wireless test system is used for testing and analysis. The monitoring results were obtained during the long-term operation of the bridge by means of accelerometers distributed by the Structural Health Monitoring (SHM) system at the quarter points of the side and main spans of the bridge. In this study, acceleration data were extracted from the SHM system on 6 February 2020 for analysis.

Furthermore, the ambient vibration test caused by excitation of random loads (e.g. wind loads, ground pulsations, water currents, etc.) in the absence of traffic loads, and acceleration data extracted from the Nanjing Qixiashan Yangtze River Bridge SHM system on February 6th, 2020 were used for the comparative analysis of the dynamic characteristics of the bridge. Comparison of the finite element simulation results with the ambient vibration test and monitoring results of the bridge is shown in Table 6, and the maximum error is only 3.2%, 1.05%.

Comment #2: What do you mean that ^As the process is smooth and random, the first natural vibration frequency of the bridge in this test was not measured, and the results are not shown in Table 6.^ ?

How do you know that 0.11 Hz is not your first frequncy from ambient vibration testing? Did you derive the mode shapes from ambient vibration tests as well?

Response: Thanks for the comment. During the ambient vibration test, the main girder test sections were arranged on the side span eight points and the middle span sixteen points, totaling 33 test sections. Through the structural ambient vibration signals picked up at each test point, the modal analysis method can be applied to identify the mode shapes at the corresponding frequencies. The following figure gives the mode shapes obtained from ambient vibration tests. Of these, of the first six orders of modes, only five were identified by the ambient vibration test.

We tried our best to improve the manuscript and made some changes in the manuscript. These changes will not influence the content and framework of the paper. And we have marked the changes in red in revised paper.

We appreciate for Editors/Reviewers’ warm work earnestly, and hope that the correction will meet with approval.

Once again, thank you very much for your comments and suggestions.
